# A Quantum Model of Trust Calibration in Human–AI Interactions

**DOI:** 10.3390/e25091362

**Published:** 2023-09-20

**Authors:** Luisa Roeder, Pamela Hoyte, Johan van der Meer, Lauren Fell, Patrick Johnston, Graham Kerr, Peter Bruza

**Affiliations:** 1School of Information Systems, Queensland University of Technology, Brisbane 4000, Australiaj.vandermeer@qut.edu.au (J.v.d.M.);; 2School of Exercise and Nutrition Sciences, Queensland University of Technology, Brisbane 4000, Australia

**Keywords:** quantum cognition, trust, artificial intelligence, probabilistic models, cognitive neuroscience

## Abstract

This exploratory study investigates a human agent’s evolving judgements of reliability when interacting with an AI system. Two aims drove this investigation: (1) compare the predictive performance of quantum vs. Markov random walk models regarding human reliability judgements of an AI system and (2) identify a neural correlate of the perturbation of a human agent’s judgement of the AI’s reliability. As AI becomes more prevalent, it is important to understand how humans trust these technologies and how trust evolves when interacting with them. A mixed-methods experiment was developed for exploring reliability calibration in human–AI interactions. The behavioural data collected were used as a baseline to assess the predictive performance of the quantum and Markov models. We found the quantum model to better predict the evolving reliability ratings than the Markov model. This may be due to the quantum model being more amenable to represent the sometimes pronounced within-subject variability of reliability ratings. Additionally, a clear event-related potential response was found in the electroencephalographic (EEG) data, which is attributed to the expectations of reliability being perturbed. The identification of a trust-related EEG-based measure opens the door to explore how it could be used to adapt the parameters of the quantum model in real time.

## 1. Introduction

The current drive for research in explainable AI (XAI) is predicated on the idea that improving predictability and understanding of the working of artificial intelligence (AI) will improve trust [1]—the idea being that if we can explain it, then we can understand it; we can know it is trustable, and trust will follow. The converse is similarly commonly assumed that, generally, people are not aware how AI technologies function and what to expect from them. Consequently, there is a low likelihood of trust [2]. These arguments are incomplete, however, as although understanding may contribute to a sense of predictability and reliability (and thereby, engender a sense of trust), this is not always the case.

One of the sources of this unpredictable quality of human–AI trust may lie in part in a distinctive aspect of human–AI interactions as opposed to human–autonomy interactions. An autonomous system is an artificial agent capable of acting independently in a dynamic environment, i.e., an autonomous system is one that makes choices on its own even when encountering uncertain or unanticipated events, often one that learns from its mistakes and changes its behaviour, and one that has beliefs and goals of its own [3]. Friedland [4] notes that AI can be distinguished as it is an artificial agent that, in addition to the qualities of an autonomous system, exhibits some aspects of what we regard as *human-like* intelligence. Jacovi et al. [5] emphasise this point, describing AI as “automation that is attributed with human-like intelligence by the human interacting with it”. Extending this positioning of AI in relation to autonomy, trust in AI can then be considered to be something like trust in autonomy combined with attributed anthropomorphised intent or reasoning. We contend that this anthropomorphised aspect is not considered to be simply trust in autonomy with an additional layer; rather, trust in AI is *imbued* with this human-like condition. Hence, we might refer to an automated system as designed to perform in a certain way and AI to have an intent to behave towards a certain goal. As a result, there can be a conceptual drift in discussing the notion of ‘machine learning’, as both a capacity for a system to react to novel situations and update its performance, as well as a sense that the AI inherently has a degree of anthropomorphised drive or intent. Such conflation can notably lead to latent assumptions of sentience and independently sustained ego projections via humans interacting with AI.

We take the view that anthropomorphism is not pre-determined by the attributes of the autonomous system or AI. That is, attributes such as the appearance of the AI (including human-like facial features or vocal attributes such as accent), behaviour, or human-like responses (for example, mimicking cultural identifiers like slang) [6] are not assumed to determine or engender anthropomorphism, although these attributes or behaviours may indeed facilitate it. Rather, we align with Jacovi et al. [5] that anthropomorphism describes when ‘human-ness’ is *attributed* to the AI via a human agent in the interactions with the system. In quantum cognition, this assumption that a cognitive property does not have a determined value or state prior to measurement is referred to as indeterminacy [7], and its significance for modelling cognitive states is discussed in more detail below.

### The Multi-Dimensional and Dynamic Nature of Trust

Before detailing our exploration of modelling trust calibration, we first note several key points regarding the nature of trust that inform this study. The first of these is that trust is taken to be multi-dimensional and holistic. By this, we mean that we adopt the well-known conception of the three dimensions of trust: reliability, integrity, and benevolence [8]. We note, however, that while each dimension is distinguishable, we hold these dimensions to be enacted holistically, i.e., the dimensions are inter-dependant and not formed in isolation of each other. Nevertheless, it is useful to consider these dimensions separately (despite being inter-related) as a means to examine trust interactions and, specifically, in order to examine human–AI trust. This study focuses on reliability as one dimension of trust.

Further, we agree that reliability can be roughly aligned with a sense of perceived ability/capability, benevolence with a sense of a common goal/shared benefit, and integrity with a sense of shared values/beliefs [9]. These alignments are asymmetric in that the human agent will prefer and judge these dimensions according to the consonance with *their* desired AI ability (‘it does the job I want it to do’), *their* goal (‘the AI will act towards helping me’), and *their* values/beliefs (‘the AI will act in a way that aligns with my values’). If the human agent and AI are highly aligned along these dimensions, they (the human agent) will experience a high degree of rapport. Trusting the AI to meet these preferences leaves the human agent vulnerable. This exploratory study focuses on human judgements of AI reliability as a dimension of trust in human–AI interactions.

The second key point is that trust is dynamic [10]. Through the interactions, the human agent’s expected and perceived trust of the AI may change, and gaps can arise between the human agent’s trust expectation and their post-measurement perceived trust in their interactions with the AI system. Although there may be a tolerance range for the disagreement of the expected and perceived trust, if the gap is perceived to be significant, then the human agent’s trust will be perturbed and may lead to re-calibration. This evolutionary aspect of trust may be particularly relevant to trust in co-operative relationships (such as when transiently coming together to undertake a task or goal that is temporarily mutually aligned) as opposed to collaborative ones (such as when teams are more stable and consolidated over repeated interactions and share a common goal). This is because co-operative relationships can be characterised as having an acknowledged potential disparity between the underlying goals of the human agent and the AI in performing the joint task. It may be that when the goals are perceived as being sufficiently aligned, that trust is afforded, i.e., is an available potentiality. This differentiation between co-operative and collaborative human–AI teaming may also be significant for trust in certain contexts (e.g., human–AI teams deployed in high-risk mission applications). In this study, we focus on the dynamic quality of trust as a cognitive state of the human agent.

To investigate a human agent’s evolving judgements of AI reliability in human–AI interactions, this paper addresses two aims. The first aim is to advance the modelling of this phenomenon by comparing the predictive performance of quantum (we use the term ’quantum’ to be inclusive of the meaning ’quantum-like’, for brevity and consistency throughout this paper) and Markov random walk models. This is initially carried out by setting the formal basis for these models, followed by presenting behavioural data from an exploratory empirical study in the form of self-report reliability ratings in a Wizard-of-Oz style human–AI experiment. The implications of these findings are later considered with regards to the assumption of indeterminacy in modelling trust. The second aim is to explore the potential for identifying a neural signature of the perturbation of a human agent’s judgement of the AI’s reliability. As this perturbation may occur unconsciously and, therefore, elude self-report techniques, a neurophysiological measure via recording electroencephalography (EEG) was added to the experimental study. The initial findings with regards to such a neural correlate in the form of event-related potentials (ERPs) are then presented. Finally, it is suggested that such a neural correlate has the potential to inform the parameters that determine the dynamics of the Markov and quantum models in future developments for modelling the calibration of trust in human–AI interactions.

## 2. Modelling of Trust

As described above, when a human interacts with an AI system, their trust expectation, including the expected reliability of the system, will fluctuate according to how they judge its performance. The problem that is addressed in this paper is how to model these fluctuations in order to accurately predict the human’s judgement of reliability. One way to view this problem is as a signal detection task.

Signal detection is a decision task in which the human agent monitors a noisy information signal for a potential target, e.g., a specialist scanning a chest X-ray for malignant tumours. In the process of signal detection, the human agent accumulates evidence for the presence of the signal. When sufficient evidence has been accumulated, the agent decides that the target is present. Conversely, at some point, if insufficient evidence has been accumulated, a decision is made that the target is not present. In this study, the noisy signal is provided by a series of interactions with an AI system, and the target being sought by the human agent is the decision of whether the AI system is reliable or not.

Random walk (RW) models are a common way of modelling the signal detection task [7]. The essential idea behind a RW model is that when a human interacts with the AI system, evidence of reliability will vary. Consequently, there will be a transition from the present level of reliability to other levels depending on how the AI conforms to the human’s expectation of reliability. For example, a human agent might hold an initial expectation of a certain level of reliability. If during a series of interactions, the AI performs as expected, evidence for this level of reliability accumulates and is confirmed. If, however, the AI responds in a way that does not conform to expectation, evidence for this level of reliability begins to dissipate and change.

### 2.1. Markov Random Walk Model

A common form of RW model is a Markov model. Reliability is assumed to be modelled on a scale from 0 (the AI is totally unreliable) to 4 (the AI is fully reliable). A Markov model assumes that at each point of time, the human agent is in a *definite* cognitive state |x〉 corresponding to one of the levels of reliability x∈{0,…,4}. As the agent interacts with the AI, their cognitive state changes in relation to the reliability that they expect. The initial probability distribution over the five states is represented by a 5×1 column vector ϕ(0), where 0 denotes time t=0. It is common to set this distribution uniformly $\phi_{x}\left(0\right)=\frac{1}{5}$, which is an approach taken when nothing is known initially about the cognitive state of the human agent being modelled. Transitions between states are governed via an intensity matrix *K*:$$\begin{array}{cc} K & =\left[\begin{array}{ccccc} -\alpha & \beta_{-} & 0 & 0 & 0\\ \alpha & -\alpha & \beta_{-} & 0 & 0\\ 0 & \beta_{+} & -\alpha & \beta_{-} & 0\\ 0 & 0 & \beta_{+} & -\alpha & \alpha \\ 0 & 0 & 0 & \beta_{+} & -\alpha \end{array}\right] \end{array}$$
The value $k_{ij}$ represents the transition intensity from row state *i* to column state *j*. For example, if a human agent inhabits a cognitive state corresponding to reliability level *i*, then $k_{ij}$ represents the intensity to transition to a cognitive state corresponding to reliability level *j*.

The non-zero off-diagonal values of *K* together with the other zero values prescribe that perceptions of a given reliability level can only transition to neighbouring states.

The parameter $\beta_{+}>0$ determines the rate of increase in expectations of reliability over time, and the parameter $\beta_{-}>0$ determines the rate of its decrease over time. The parameter $\alpha =(\beta_{+}+\beta_{-})$ represents the level of intensity for the human agent to remain at the current level of expectation regarding reliability.

The transition matrix obeys the Kolmogorov forward equation:$$\begin{array}{cc} \frac{d}{dt}T\left(t\right) & =KT(t),\\ T(t) & =e^{tK} \end{array}$$

The probability distribution over states at time *t* is similarly defined:$$\begin{array}{cc} \frac{d}{dt}\phi \left(t\right) & =K\phi (t),\\ \phi (t) & =e^{tK}\phi \left(0\right) \end{array}$$
Markovian dynamics describe the probability of transitions between cognitive states that are definite. More formally, the probability distribution is assumed over reliability ratings at time *t* is given by $\phi \left(t\right)={\left[p_{0},p_{1},p_{2},p_{3},p_{4},\right]}^{T}$. This distribution formalises the assumption that the probability that the cognitive state has the reliability expectation equal to *i* is given via $p_{i}$, 0≤i≤4. Therefore, the prediction that the reliability rating *R* is equal to *i* at time *t* is given via
(1)$$\begin{array}{cc} P_{t}\left(R=i\right) & =p_{i} \end{array}$$

In summary, Markovian dynamics can be envisaged as follows: Each interaction with the AI transitions the current cognitive state corresponding to a reliability rating to another definite cognitive state corresponding to the same or neighbouring reliability state, depending on the nature of the interaction. Consequently, interactions with the AI system are modelled as a discrete path through the space of possible cognitive states.

### 2.2. Quantum Random Walk Model

A quantum RW model (henceforth “quantum model”) [7] assumes that prior to being asked to rate the level of reliability of the AI, the human agent is in an indeterminate cognitive state regarding the levels of reliability. This is in contrast to the Markov RW model, which assumes the human agent is in a definite cognitive state at all times. One way to understand this quantum indeterminacy is that before the human agent is asked to rate the reliability of the AI, the cognitive state is a superposition of *all* levels of reliability. When the human agent is asked to decide on a reliability rating, the superposition collapses into a definite cognitive state corresponding to a single reliability level, and then, this level is what is reported. In other words, before being measured, all levels of reliability coexist as propensities in the cognitive state. Such propensities are not yet actual, and hence, the cognitive state is indefinite.

The quantum RW model is based on five orthonormal basis vectors |x〉 for x∈{0,…,4}, which span a 5-dimensional real Hilbert space. The initial cognitive state of the human agent is assumed to be superposed across all of the five reliability levels, $|S(0)\rangle =\sum_{x}\psi_{x}\left(0\right)|x\rangle$, which is a linear combination of the five orthonormal basis vectors. The superposition state evolves continuously across time until the human agent is asked to supply a reliability rating based on their interactions with the AI.

In contrast to Markovian dynamics, the quantum dynamics are analogous to a wave-like evolution of a cognitively superposed state [7]. Consequently, the quantum dynamics are very different to Markovian dynamics. The differentiating factor is the assumption of whether the cognitive state is always definite (Markov) or whether it is indeterminate (quantum). It is the indeterminate, “wave-like" nature of the superposed cognitive state that allows multiple state transition paths to be taken at once through the cognitive state space. The wave collapses onto a definite cognitive state when a human agent is asked to record a reliability rating.

The dynamics of the quantum model are prescribed by a n×n Hamiltonian matrix *H*, where *n* is the dimensionality of the Hilbert space. For example, when n=5,
$$\begin{array}{cc} H & =\left[\begin{array}{ccccc} \mu_{0} & \sigma^{2} & 0 & 0 & 0\\ \sigma^{2} & \mu_{1} & \sigma^{2} & 0 & 0\\ 0 & \sigma^{2} & \mu_{2} & \sigma^{2} & 0\\ 0 & 0 & \sigma^{2} & \mu_{3} & \sigma^{2}\\ 0 & 0 & 0 & \sigma^{2} & \mu_{4} \end{array}\right] \end{array}$$
The value in cell $h_{ij}$ specifies the diffusion of the amplitude of the wave from column state *j* to row state *i*. Therefore, the parameter $\mu_{x}$ specifies the diffusion rate back into a basis state |x〉. In other words, it corresponds to energy for the human agent to keep expectations of reliability at certain level in relation to a reliability rating *x*. The diagonal values correspond to the nature of the potential function, which drives the dynamics. When the diagonal values are increasing, e.g., $\mu_{x}=\beta x$ specifies that a constant positive force is being exerted, which drives the energy of the wave towards higher reliability ratings *x*. This occurs when the AI system is conforming to expectations. In contrast, the parameter $\sigma^{2}$ specifies the diffusion of wave amplitude away from a given reliability state to neighbouring reliability states. The non-zero off-diagonal values correspond to wave energy dissipating to neighbouring reliability levels, which is the quantum equivalent of the off-diagonal values on the Markov transition matrix *K*.

In addition to the standard quantum RW Hamiltonian (which sets all other cells $h_{ij}$ to zero), we introduced a parameter *z* to explore the diffusion of expectations of reliability at the extremities of the reliability ratings. The cells, $h_{15}=z$ and $h_{51}=z$ specifies a diffusion of wave energy from the highest reliability rating to the lowest. This is intended to model the situation when expectations of high reliability are perturbed, e.g., when the AI system suddenly does not conform to expectations, and consequently, a low level of reliability starts to permeate the dynamics. The *z* parameter perturbs the wave-like dynamics of the quantum model to make them more uneven. As Markov dynamics are not wave-like, we did not deem it useful to introduce this parameter to the Markov RW model.

The dynamics of the quantum model are defined via a unitary matrix using the Schrödinger equation:$$\begin{array}{cc} \frac{d}{dt}U\left(t\right) & =-iHU(t)\\ U(t) & =e^{-itH} \end{array}$$
The evolution of the cognitive state ψ at time *t* is given via:$$\begin{array}{cc} \frac{d}{dt}\psi \left(t\right) & =-iH\psi (t),\\ \psi (t) & =e^{-itH}\psi \left(0\right) \end{array}$$

Note the distinction between the Markovian and quantum dynamics: The quantum cognitive state ψ can be viewed as a wave which has amplitudes at the different reliability ratings. The square of the amplitude of the wave at a given reliability rating corresponds to the probability that the human agent will record that reliability rating when asked at a time *t*. More formally, assume the cognitive state at time *t* is given by $\psi \left(t\right)={\left[q_{0},q_{1},q_{2},q_{3},q_{4},\right]}^{T}$ where $q_{x}$ is the amplitude of the wave associated with the corresponding cognitive basis states: |x〉∈{0,…,4}. The prediction that the reliability rating *R* is equal to *i* at time *t* is given via
(2)$$\begin{array}{cc} Q_{t}\left(R=i\right) & =q_{i}^{2} \end{array}$$
where $Q_{t}\left(\cdot \right)$ denotes a quantum probability.

In short, quantum dynamics prescribe how an indeterminate cognitive state evolves through time like a wave. The quantum and Markov models are illustrated in Figure 1.

## 3. A Wizard of Oz Human–AI Experiment

In order to examine comparable applications of the Markov and quantum RW models to human agents judging AI reliability, we undertook an empirical study in the form of a Wizard-of-Oz (WoZ) style experiment collecting self-reported reliability ratings. WoZ refers to a person(s), usually the experimenter(s), who remotely operates machine intelligence [11]. WoZ may involve “any amount of control along the autonomy spectrum, from fully autonomous to fully tele-operated, as well as mixed initiative interaction”.

As noted above, in addition to modelling the calibration of the reliability signal over time, we are interested in the interaction of when reliability is perturbed. As this perturbation may be quite nuanced or subtle and perhaps occur unconsciously, a neuro-physiological measure via recording EEG was added to this study. As noted by Kohn et al. [12], “the application of neural measures to trust is relatively new and somewhat exploratory”, especially in research in trust in automation and AI. However, if such a neuro-physiological indicator can be reliably identified, there is potential for this event to inform the modelling parameters that determine the dynamics of the Markov and quantum RW models.

In the experiment, human participants were required to interact with an AI system based on an image classification task. Specifically, participants were shown images of human faces one by one and asked to judge whether it was real or fake (that is, AI generated). After the participants recorded their judgement, they were then shown the AI classification of the image as real or fake. The Wizard-of-Oz element is that the responses of the AI system were manipulated to often agree with the participant’s response. This is intended to promote the expectation that the AI is reliable. Occasionally, the AI system disagrees with the human response, thereby perturbing the expected reliability that has been promoted.

At regular intervals, participants rated the reliability of the AI system, and these ratings data form the baseline to measure the predictive performance of the Markov and quantum models. In addition, electrical brain activity was recorded via EEG. These data were analysed to identify potential neural correlates of when reliability expectations have been perturbed. Other ancillary data were also collected, e.g., reaction times, participant image classification performance, participants’ subjective attitudes to their own performance and the AI’s performance, and generalised trust in AI both before and after the experiment.

In order to address the current aims of this exploratory study, to compare the predictive performance of quantum vs. Markov RW trust models and explore the potential to identify a neural correlate of trust perturbation of a human interacting with AI, in this paper, we focus on the first results of the reliability ratings and event-related potential (ERP) analyses of the EEG signals.

### 3.1. Participants

The participants were healthy adults (no eye, musculoskeletal, neurological, or psychiatric diseases/injuries) between 18 and 30 years of age, and they received financial compensation in the form of a non-specific gift voucher (AUD 20). In this exploratory study, seven participants took part. The participants were instructed before and during the experiment that they were interacting with an AI ‘agent’. Only after the experiment were they debriefed to clarify that the ‘AI agent’ was a pre-programmed Wizard of Oz (WoZ).

### 3.2. Materials

We used the human-face image set from Nightingale and Farid [13] for this current study. From this set, 294 AI-synthesised (fake) faces and the same number of real faces were selected (that is, 50% real face images and 50% fake). The selected images ensured diversity across gender, age, and ethnicity. In addition, we selected the top 294 images in each category (real or fake) based on the accuracy rating in Nightingale and Farid [13]; that is, we selected the images that were most correctly identified as real or AI-synthesised by human participants in Nightingale and Farid [13] in order to better manipulate the perception of the (WoZ) AI system’s reliability.

The order of images was randomised across trials, blocks, and participants, and each image was displayed once only (no repeats). Participants classified the images using the left and right arrow keys on a standard keyboard. Half of the participants had the left-key assigned for the real rating and the right-key for fake, and vice versa for the other half of participants (in counterbalanced order).

### 3.3. Experimental Protocol

The experiment comprised 20 blocks of 28 trials each (560 trials in total), as well as a practice block with 28 trials at the start. A visual overview of the experimental design is given in Figure 2, in which Figure 2B depicts the trial structure. Each trial started with a fixation cross in the centre of the screen between 0.6 and 1 s. Subsequently, the image stimulus was displayed until the participant pressed a response key (classification of real or fake) or for a maximum of 2 s. Upon pressing a response key, their own classification was displayed on the screen: either a blue human icon if they classified the image as real or a yellow human icon if they classified the image as fake. A short period after the human icon display (0.3 s), the AI classification was also displayed on the screen: either a blue robot icon if classified as real, or a yellow robot icon if classified as fake. Both the human and robot icon remained on the screen for 1 s before the next trial began.

After each block (as well as before and after the practice block), participants were asked to rate the reliability of the AI classification on a continuous scale with a slider from 0 to 100 using the mouse (Figure 2C). Completing the reliability rating was self-paced. The participant response was then displayed on the screen for 2 s, and the participant could not change their response after it was first logged. After completion of the reliability slider, a message on the screen invited the participants to take a short self-paced break before continuing when ready.

The experiment contained two conditions (Figure 2A): match and mismatch. In the match condition, the AI classification matched the participant’s classification response. That is, if the participant classified an image as real, the AI agent also classified the image as real, and vice versa; if the participant classified an image as fake, the AI also classified it as fake. In the mismatch condition, the AI classification is always different from the participant’s classification. That is, if a participant classified an image as real, the AI classified it as fake (and vice versa if a participant chose fake, the AI classified it as real). Within each block of 28 trials, 75% were the match condition, and 25% of trials were the mismatch condition. The order of match and mismatch trials was randomised in each block.

The experiment was run on a 64-bit Microsoft Windows 10 PC. The images and classification icons were displayed against a grey background (rgb [128, 128, 128]) using the PsychoPy3 standalone software version 2022.1.4 [14] onto a monitor with a 1920×1080 pixels resolution and a 60 Hz refresh rate, located 60 cm away from the participant with the centre of the screen at eye height. Images were scaled not to exceed 6° of visual angle in the vertical or horizontal direction, resulting in an image size of 6×6 cm on the screen.

### 3.4. Data Acquisition

Electroencephalographic (EEG) data were recorded using a 64-channel actiCAP system with active electrodes (BrainProducts GmbH, Gilching, Germany) placed according to the extended international 10–20 system [15], with FCz as the recording reference and Fpz as the ground.

EEG signals were recorded continuously using the BrainProducts LiveAmp amplifier and BrainVision Recorder software (version 1.25.0101), with a 500 Hz sampling frequency on an independent recording PC running Windows 10. The triggers from the PsychoPy PC were sent to the EEG system via USB port that mirrors a parallel port and the BrainProducts TriggerBox and Sensor and Trigger Extension. At the beginning of the session, the experimenter fitted the participant with the EEG cap. Recording sites on the scalp were abraded using the Nuprep paste. EEG electrodes were attached onto the cap, filled with conductive gel (Supervisc, BrainProducts GmbH, Germany), and adjusted until impedances were below or close to 10 kOhm. Before and during the experiment, the participant was instructed to relax shoulders, jaw, and forehead and to minimise swallowing.

### 3.5. Data Analysis

EEG data were pre-processed in MATLAB (version 2021b) and EEGLAB (version 2022.1) [16] using custom written routines. We down-sampled the data to 250 Hz to improve computational processing times. In brief, we followed these steps: (1) The EEG data were first high-pass filtered at 1 Hz (Hamming windowed sinc FIR filter using the pop_eegfiltnew function implemented in EEGLAB). (2) Line noise (50 Hz) was estimated and removed using the Cleanline tool [17,18]. (3) Epochs and channels with excessive noise (large-amplitude movement artefacts, EMG activity, electrode pops) were removed from the data via visual inspection. (4) Ocular artefacts were identified and removed with independent component analysis (ICA) and ICLabel [19]. For ICA, we used the extended infomax ICA algorithm with PCA option. (5) Artefact subspace reconstruction (ASR) was performed to automatically reject bad portions of the data (such as irregular large movement artefacts or bursts) [20]. The threshold for burst removal was set to 10 (standard deviations of PCA-extracted components). (6) EEG signals were re-referenced to a common average reference. (7) ICA was run again, and muscle and other residual artefacts were removed based on the visual inspection of the topoplots and the power spectra of the components. We retained all brain components and projected them back into channel space.

At each step of this pipeline, when an artefact was identified (in steps 2, 3, 4, 5 and 7), the artefact itself was extracted from the data and saved to a disk. Subsequently, we subtracted these artefacts from the raw EEG data, which yielded a clean dataset that was not high- or low-pass filtered to retain both high- and low-frequency components of interest. This approach is similar to Bigdely-Shamlo et al. [17]. Finally, channels that were removed during the cleaning process were interpolated.

After the EEG signals were cleaned, we conducted event-related potential analyses (ERP); for these analyses, we used the ERPLAB toolbox [21]. The main event of interest for our ERP analyses is the event when the AI feedback icon (robot head) is displayed on the screen. Therefore, we segmented the pre-processed EEG data into trials relative to this event from 1300 ms before to 950 ms after in order to identify the ERPs time-locked to this event. A baseline correction was applied based on 600 ms of the fixation cross period preceding the image stimulus display. Trials with large voltage changes (>100 μV) were removed from all ERP analyses, as well as trials in which participants did not press the response key in time (>2 s). For each participant, we calculated the ERPs over each electrode and also averages across frontal, central, left and right temporal, parietal, and occipital regions per condition (match or mismatch) and the difference ERP between conditions (match–mismatch).

## 4. Results

For the purposes of this exploratory paper, we present the results of one representative participant. All data are securely stored on an internal server and will be processed as this research program develops further.

### 4.1. Markov and Quantum Models of Reliability Ratings

Both the Markov and quantum models have n=11 dimensions with |x〉∈{0,1,…,10}. This choice was made to evaluate coarser grained models than 0…100 dimensionality of the scale used to capture the participant reliability ratings. In this way, we aimed to strike a compromise between higher dimensionality and model simplicity in order to better analyse dynamical behaviour.

Reliability ratings were mapped to model dimensions as follows: Reliability ratings below 9% were mapped onto x=0, ratings between 9% and 17.99% onto x=1, between 18% and 26.99% onto x=2, …, and ratings ≥90% onto x=10.

We did not perform any formal model fitting and comparison to optimise parameters as part of this exploratory study. Here, we present the modelling results visually in the form of figures alongside root mean square error (RMSE) estimates of the models as numerical indicator of the goodness of fit.

#### 4.1.1. Definition of the Initial State Vectors

The initial cognitive state vector (ϕ(0) for Markov, ψ(0) for quantum) is represented by a 11×1 vector that gives the initial probability distribution (at time t=0) over all 11 states. As stated previously, this state vector conventionally represents as uniform probability distribution. As part of this study, we tried another approach by weighting and distributing the probabilities across all 11 states non-uniformly. This is explained in detail below, and the results of the different initial state vector definitions and their effect on the Markov and quantum models is shown in Figure 3.

The non-uniform initial state vectors (ϕ(0) for Markov, ψ(0) for quantum) were defined via the symmetric distribution of probabilities centred around the participant’s rating of the AI’s reliability after the practice block. For example, if a participant gave a reliability rating of 48.9%, it was mapped onto state x=5.

For the Markov RW model, we set the probabilities of the 11 states centered around this identified start state (e.g., x=5), with peak probability at *x* and with a width of the distribution across seven states (the three neighbour states below and above the start state *x*, e.g., from states 2, …, 8). Similar to a Gaussian distribution, the start state *x* was mapped with the highest probability, and the neighbour states had decreasingly lower probabilities; the states beyond the three nearest neighbour states were assigned with zero probability (e.g., state 1 or state 10). For instance, if start state x=5, then
$$\phi {\left(0\right)}^{⊤}=\left[0,0,0.025,0.0625,0.17,0.5,0.17,0.0625,0.025,0,0\right]$$

For the quantum RW model, the initial superposed cognitive state is specified using the state vector ψ(0). Similar to how we defined ϕ(0), we set the probability amplitudes across the 11 states to be symmetrically distributed around the start state *x* of the participant with a width of seven states. For instance, if start state x=5, then
$$\psi {\left(0\right)}^{⊤}=\left[0,0,0.17,0.24,0.41,0.71,0.41,0.24,0.17,0,0\right]$$
Note that these values represent wave amplitudes.

#### 4.1.2. Definition of the Markov Intensity and Quantum Hamiltonian Matrices

The parameter settings for the Markov intensity matrix *K* were set as $\beta_{-}=0.4$ and $\beta_{+}=0.6$, which leads to α=1.0.

For the Hamiltonian *H*, the potential function used $\mu_{x}=0.05x$ and $\sigma^{2}=0.7$. The idea behind this positive function was to model the contrived conformance of the AI system to human expectations for the majority of trials, thus generating positive wave energy towards higher reliability ratings, which, in our study, is assumed to accumulate linearly. The value of $\mu_{x}=0.05$ is a scalar that prescribes the speed of accrual of this energy. Various values of $\mu_{x}$ were investigated, and a value of 0.05 was found to provide the basis of fairly good predictions, particularly during the early phases of the experiment when the participant tends to have higher levels of uncertainty about the AI system’s reliability.

After exploratory studies with z=0, setting z=0.2 seemed to adequately balance accrual of wave energy towards higher reliability ratings due to the WoZ AI often agreeing with the participant’s classification of faces vs. perturbation when the WoZ AI disagreed with the participant’s classification.

In summary, the Hamiltonian *H* used in the analysis below is defined as follows:$$\begin{array}{cc} H & =\left[\begin{array}{ccccccccccc} 0 & 0.7 & 0 & 0 & 0 & 0 & 0 & 0 & 0 & 0 & z\\ 0.7 & 0.05 & 0.7 & 0 & 0 & 0 & 0 & 0 & 0 & 0 & 0\\ 0 & 0.7 & 0.1 & 0.7 & 0 & 0 & 0 & 0 & 0 & 0 & 0\\ 0 & 0 & 0.7 & 0.15 & 0.7 & 0 & 0 & 0 & 0 & 0 & 0\\ 0 & 0 & 0 & 0.7 & 0.2 & 0.7 & 0 & 0 & 0 & 0 & 0\\ 0 & 0 & 0 & 0 & 0.7 & 0.25 & 0.7 & 0 & 0 & 0 & 0\\ 0 & 0 & 0 & 0 & 0 & 0.7 & 0.3 & 0.7 & 0 & 0 & 0\\ 0 & 0 & 0 & 0 & 0 & 0 & 0.7 & 0.35 & 0.7 & 0 & 0\\ 0 & 0 & 0 & 0 & 0 & 0 & 0 & 0.7 & 0.4 & 0.7 & 0\\ 0 & 0 & 0 & 0 & 0 & 0 & 0 & 0 & 0.7 & 0.45 & 0.7\\ z & 0 & 0 & 0 & 0 & 0 & 0 & 0 & 0 & 0.7 & 0.5 \end{array}\right] \end{array}$$

### 4.2. Event-Related Potentials

As stated, the second aim of this study was to explore whether a neurophysiological measure for trust perturbation can be identified. As shown in Figure 4 we found a clear ERP response over central cortical regions, while both conditions (match and mismatch) elicit an increase in ERP amplitude at approximately 380 ms after the display of the AI feedback icon on screen, the ERP amplitude is notably larger in the mismatch condition than in the match condition. In addition, at approximately 270 ms after the display of the AI feedback icon there is a local minimum in the mismatch condition ERP only (see Figure 4A). At all other time points, the ERP response is fairly similar for both conditions. The topographies (Figure 4B) show that the difference ERP (difference between conditions) is greatest over central regions, especially at 268 ms and 376 ms post AI feedback display.

## 5. Discussion

### 5.1. Predictive Modelling of Reliability Ratings

The initial state vector ϕ(0) (Markov) or ψ(0) (quantum) is an important modelling parameter, which greatly influences the predictions of cognitive states over time, particularly in the quantum model. A uniform probability distribution across states in ϕ(0) and ψ(0) and a weighted Gaussian-like distribution were explored (see Figure 3).

An initial state vector with uniform probability distribution (Figure 3A) yields a fairly regular ’wave’ of the quantum predictions across the experiment based on the center of the defined cognitive states (i.e., x=5 with x∈{0,1,…,10}). Notably, with a uniformly distributed initial state vector, the predictions are the same for different participants, regardless of the reliability ratings they express at the beginning of the experiment.

To address this, we found using an initial state vector with weighted probabilities and taking into account the initial cognitive state of the participant based on their first reliability rating (Figure 3B) appears to lead to a better model fit for both quantum and Markov models. That is, the predictions shown in Figure 3B appear to better align with the observed variation of the reliability ratings than in Figure 3A. For example, if a participant gives a reliability rating of 80% at the beginning of the experiment, the quantum predictions are closer to the expressed cognitive state of x=8. Moreover, an initial state vector with weighted probabilities renders more ’irregular’ quantum dynamics across the experiment than a uniformly distributed one. We note that while we provide RMSE figures for reference, as a global value they are limited. The reason for this is that different phases of the human reliability ratings across the experiment tend to favour each of the models. More specifically, in the initial phase of the experiment, the reliability ratings tend to oscillate for a period, which is more amenable to modelling via quantum dynamics. In contrast, the phase thereafter appears to show the oscillations in reliability dampening and evening out, which is more amenable modelling with a Markov model.

Moreover, Figure 3 demonstrates that the definition of the initial state vector greatly influences the Markov and quantum predictions. The visual inspection of the predictions (Markov and quantum) and experimental observations shown in Figure 3B,C compared to Figure 3A highlight that an initial state vector with weighted probabilities tends to lead to a better fit of the experimental data than an uniformly distributed initial state vector. The dynamics of the Markovian predictions seem to be somewhat less strongly influenced by the initial state vector than the quantum predictions. However, an initial state vector with weighted probabilities (Figure 3B) also seems to improve Markovian predictions as the curve at the earlier time points is steeper compared to the curve based on an initial state vector with uniformly distributed probabilities (Figure 3A).

It is striking how greatly the initial state vector influences the evolution of the quantum dynamics throughout the entire experiment. It does not only change the predictions at early time points, but it affects all time points, and it influences the shape of the peaks in the wave (for example, the amplitudes of the peaks are flatter in Figure 3B compared with A). Overall, our exploratory investigations to date highlight that the predictive performance of the quantum model is sensitive to the initial state vector. However, we acknowledge that our observations are exploratory in nature, and additional analysis is required to understand the effect of the initial state vector on the subsequent dynamics.

The Markovian dynamics are influenced by the parameters $\alpha ,\beta_{+}$ and $\beta_{-}$ in the intensity matrix *K*. Despite exploring various combinations of parameters, it was not possible to find parameters to fit the fluctuations in reliability ratings, especially in the early stages of the trials. At later stages of the experiment, when a participant tends to fluctuate less in their reliability ratings, the Markovian predictions seem to fit the experimental data points better. We will return to this point in more detail below.

The quantum dynamics are influenced by the parameters $\mu ,\sigma^{2}$ and *z* of the Hamiltonian *H*. Various parameter settings were explored, and it was easier to find parameters that fit the fluctuations of the reliability ratings, as quantum dynamics are wave like, whereas Markovian dynamics are not [7]. We found that introducing the parameter z=0.2 at $h_{1n}$ and $h_{n1}$ of the Hamiltonian *H* created more ‘bumpy’ and irregular quantum dynamics across all time points, improving the quantum predictions further compared to z=0 (Figure 3B,C). We conclude that a non-zero *z* parameter has potential for providing a better fit to the experimental data as it allows wave energy to diffuse between non-adjacent cognitive basis states. For example, a human agent’s assessment of an AI’s reliability may be brittle, which can lead to a drastic re-assessment based on a single interaction with the system and, thus, lead to transitions between non-neighbouring cognitive states. This contrasts with the incremental adjustments of reliability specified using RW dynamics. Further research is needed to understand the affect of this parameter in order to better tune it. It is possible that multiple *z* parameters in the Hamiltonian might be needed.

Determining the optimal parameters for a model is challenging, regardless of whether it is quantum or Markov. We envisage an adaptive approach that combines real-time analysis of EEG to influence the dynamics of the models using the results of this analysis to update the parameters in the Hamiltonian *H* or Markov transition matrix *K*.

### 5.2. Determinate vs. Indeterminate Trust

In the introduction, we proposed that trust is a cognitive property that is indeterminate prior to measurement. This assumption contrasts a commonly held view in cognitive psychology [7,22], namely that, at all times, trust corresponds to a determinate cognitive property and that the underlying definite value of that property is simply captured and reported at the time a human agent is asked to supply a rating.

Although the preceding analysis is exploratory in nature, the assumption of indeterminacy does seem to provide a promising dynamical model that can better predict the fluctuations of reliability (our proxy for trust), particularly in the early interactions with the AI. One of the purported strengths of quantum dynamics is its ability to account for systematic oscillation effects across time [23].

An observed pattern encountered across some participants is that the early fluctuations in the judgements of reliability tend to dampen as the number of interactions increases. Such behaviour has been encountered in modelling changes in confidence levels [24], preferences [25], and ambivalence in decision making [26]. For this reason, open quantum systems have been proposed as potentially useful theory to develop quantum decision models [23,27]. Open quantum systems were developed to formalise how quantum systems interact with the environment. Such quantum systems exhibit fluctuations that dampen as time progresses because their inherent indeterminacy is dissipating towards a classical system in which all states are determinate. By way of analogy, the fact that the fluctuations in reliability rating tend to decrease with the number of interactions with the AI system (the environment) suggests that underlying cognitive state of trust is becoming more “classical" as time progresses.

An advantage of the open quantum systems approach is that it is theoretically principled because it computes a *single* probability distribution from a unified process containing a quantum component, which models the initial oscillatory dynamics and later converges to a Markov component that ultimately reaches equilibrium. This is preferable over an ad hoc hybrid model, which simply averages the two components, because in resulting dynamics, both are always operating and present [27]. As illustrated in Figure 3, this is clearly not the case. The initial phase is quantum, followed by a Markovian phase.

A second advantage is that open quantum systems allow two different types of uncertainty to be combined. The first is epistemic uncertainty, which covers the modeller’s lack of knowledge regarding the cognitive state of the decision maker. The second is the indeterminacy of the underlying cognitive state. This has been referred to as ‘ontological uncertainty’ because the uncertainty pertains to the very nature of the cognitive state itself, not lack of knowledge about it [28].

In summary, assuming that the underlying cognitive state of trust is indeterminate opens up new theoretical perspectives and modelling approaches to advance the understanding of trust in human–AI interactions.

### 5.3. Potential Neural Correlate of Trust Perturbation

The large difference in ERP amplitude between match and mismatch conditions, especially at latencies between 300 and 500 ms, may be indicative of a P3 response. The P3 wave describes a positive voltage of an EEG signal between approximately 300 to 600 ms after the relevant stimulus (in this study, the display of the AI feedback icon at time 0 in Figure 4) and is usually elicited by infrequent stimuli [29,30]. A P3 wave is commonly observed in oddball paradigm ERP studies [29,30]. The main characteristic of oddball paradigms is that they test and compare frequent and rare stimuli. For example, in this study, the match condition is frequent (75% of trials), and the mismatch condition is rare (25% of trials). In general, the P3 amplitude is larger for lower target probabilities [29,30]. In other words, the P3 amplitude is larger for the rare stimulus category than for the frequent stimulus category, which our results agree with. Hence, in the context of this study, we may conclude that the P3 response may represent a neural correlate of perturbed trust. However, this interpretation is noted with caution as it needs to be confirmed in further participants.

ERP responses at latencies between 0 and 200 ms after a stimulus represent sensory responses (e.g., visual processing), and later latencies are mostly related to higher-order cognitive processing [30]. As the difference in P3 amplitude between rare and frequent stimuli conditions starts to appear at latencies beyond 300 ms, it is thought that neural processes related to stimulus categorisation (the process of the brain to categorise stimuli) must be completed before the onset of the P3 wave [30].

In summary, we found that the ERP results provide support for the potential of EEG-based measures to supply a neuro-physiological signature of perturbations of reliability judgements during human–AI interactions and, therefore, of trust more generally.

In the next step, we will need to confirm these ERP responses in multiple participants and extract features from the ERP waves to be used for the quantum and Markov modelling. In particular, we will explore ERP features to update parameters of the Hamiltonian and Markov intensity matrix in real time. We expect that adding EEG-derived variables into the models will improve predictions of participants’ reliability ratings and fit underlying assumptions of open quantum systems better. We suggest that such a step is necessary for the real-time modelling of trust the human–AI interaction, a future goal of this research.

## 6. Conclusions

Quantum cognition has been successful in delivering models that systematically account for how humans transact decisions ‘on the ground’, rather than what decisions they *should* make. This article presents an exploratory study of human judgements of reliability when interacting with an AI system. A quantum random walk model was shown to have potential in predicting the reliability ratings of individual participants, particularly in the earlier phases of the interactions when uncertainty is high, and these ratings tend to fluctuate but were less successful in the later phases of the interactions when participant ratings tend to stabilise. More participants will need to be studied to determine the prevalence of this pattern. If this pattern is confirmed across several participants, an open quantum systems approach is worth investigating to account for this pattern. The open quantum systems approach is based on the Linblad master Equation [31].

Future research could be directed at determining how the event-related potentials corresponding to the perturbation of a participant’s expectation can be used to inform the specification of Linblad operators to allow the dynamics to be adapted based on a real-time analysis of EEG data. This would open the door to the development of frontier neuro-quantum technologies that would facilitate the real-time prediction of a human agent’s trust in their interactions in AI. The ambitions of this type of research is that these predictions could allow for interventions when the trust in AI is predicted to be low. The ultimate goal of such technologies is to promote effective trusted interactions between human and machine intelligence.

## Figures and Tables

**Figure 1 entropy-25-01362-f001:**
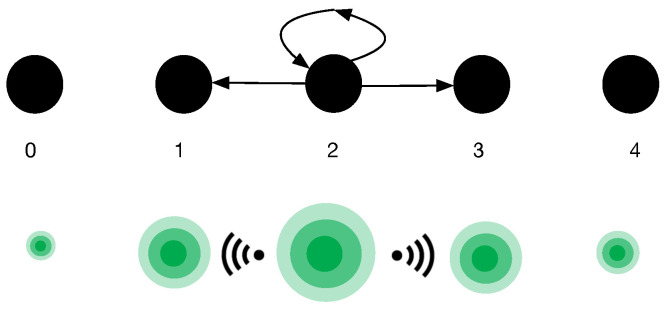
Markov and quantum models. The top depicts a Markov model. The black dots represent definite cognitive states associated with a reliability rating. The human agent currently inhabits the cognitive state associated with a reliability rating of 2. In a RW walk model, there can be a transition to adjacent states or back to the current state. Probabilities are associated with these transitions. The quantum model (bottom) illustrates that the human agent simultaneously inhabits all cognitive basis states which are like wave peaks. The concentric circles represent the amplitude of the wave peaks at each basis state, which represent probabilities. Wave energy dissipates to neighbouring wave peaks.

**Figure 2 entropy-25-01362-f002:**
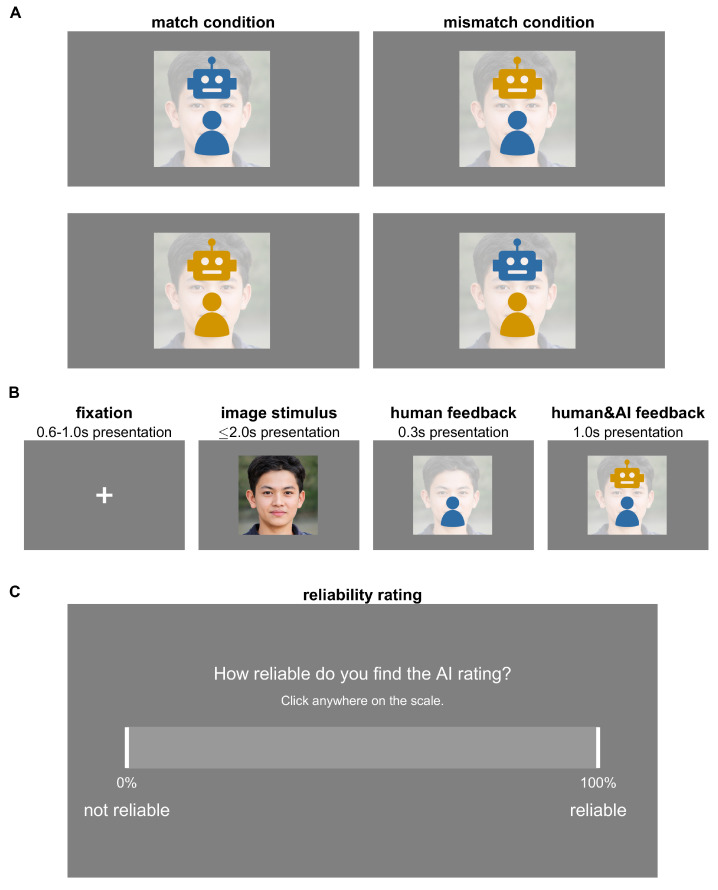
Overview of the experimental design. (**A**) Two conditions: match and mismatch. In the match condition human and AI feedback icons have the same colour, indicating agreement on the image classification (both judge image as ‘real’ or ‘fake’). In the mismatch condition, the human and AI feedback icons have different colours, indicating disagreement. Blue indicates a ‘real’ classification, and yellow indicates a ‘fake’ classification of the image. (**B**) General sequence of each of the 560 experimental trials: display of a fixation cross on grey background for 0.6–1.0 s (variable and random duration), display of image until the participant makes their classification choice (real or fake) or for a maximum of 2.0 s, display of the participant’s classification choice (human icon at the bottom of the screen) for 0.3 s (if participant did not make a classification choice, no feedback was displayed), and display of human and AI classification (human icon at the bottom and robot icon at the top of the screen) for 1.0 s. (**C**) Reliability rating: after a block of 28 trials, behavioural ratings of the reliability of the AI image classification are collected (self-paced). The image shown in this figure is from Nightingale and Farid (2022) (public domain).

**Figure 3 entropy-25-01362-f003:**
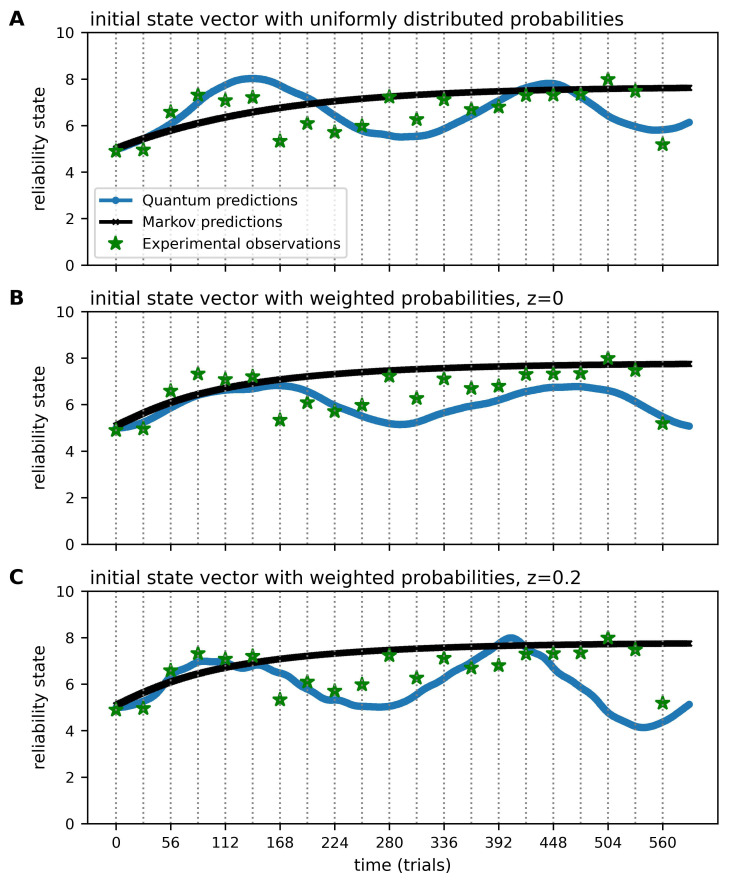
Evolution of human cognitive states with respect to reliability of the AI image classification performance showing experimental, Markov and quantum random walk predictions. The initial state vector ϕ(0) for the Markov and ψ(0) for the quantum predictions was uniformly distributed (**A**) or symmetrically weighted ((**B**,**C**); see text for details). The definition of the Hamiltonian *H* included z=0 (**B**) and z=0.2 (**C**). The x-axis shows the number of trials with grey, dotted vertical lines indicating each block of the experiment (28 trials per block), and the y-axis shows the (observed or predicted) cognitive state of the participant. In (**A**), the Markov RMSE is 0.92 and quantum RMSE 0.97; in (**B**), the Markov RMSE is 1.0 and quantum RMSE 0.92; in (**C**), the Markov RMSE is 1.0 and quantum RMSE 1.28. Figure shows data of one representative participant.

**Figure 4 entropy-25-01362-f004:**
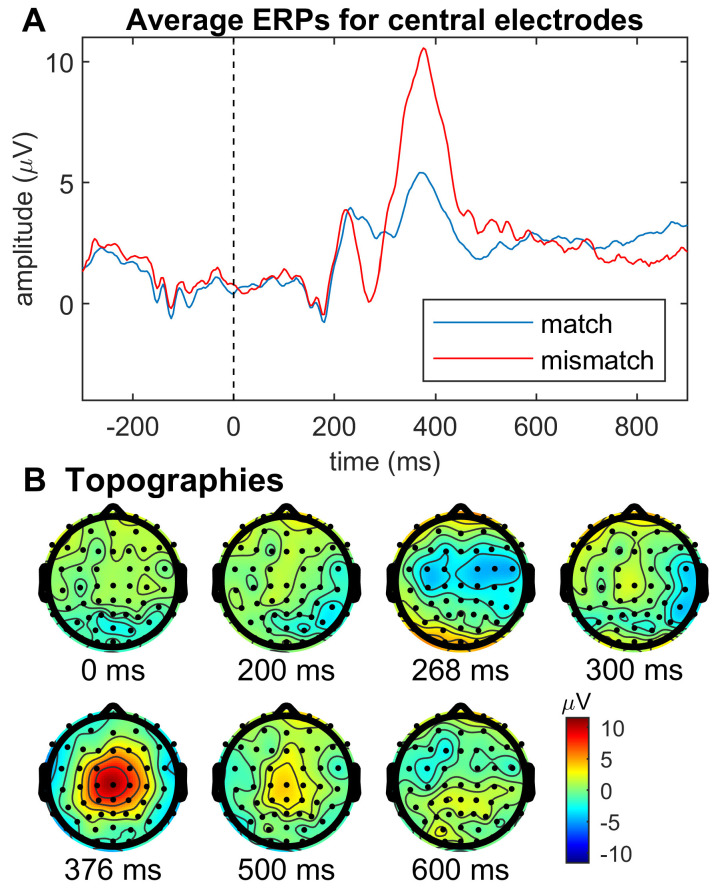
(**A**) Time course of event-related potentials (ERPs) time locked to the event when the AI rating icon (robot head) is displayed on the screen (0 ms) averaged over central EEG electrodes in the matched (human and robot icon same colour, i.e., both rate fake or both rate real) and mismatched (robot icon colour is opposite to human colour choice, i.e., human = blue and robot = yellow or vice versa) conditions. Waveforms represent the mean ERP across all trials for each condition of one representative participant. (**B**) Topographies of the difference ERPs at various time points at/after the AI rating icon display event (difference ERP refers to ERPmatch–ERPmismatch). Time points correspond to time course shown in A. For instance, at time point 376 ms, post AI feedback icon display is a local maximum in both match and mismatch ERPs, albeit of greater amplitude in the mismatch condition. The topographical plot in B shows peak activation over central electrodes.

## Data Availability

The anonymised data presented in this study are available upon reasonable request from the corresponding author.

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
