# Peer review of "A Quantum Model of Trust Calibration in Human–AI Interactions"

_entropy, 2023, doi:10.3390/e25091362_

Round 1
Reviewer 1 Report
This is an interesting paper and I enjoyed reading it. The results are clearly very preliminary but are nevertheless intriguing. I think it is worthy of publication, but I do have three minor suggestions for clarifications that I think would be useful for the reader:
1) The paper contrasts a quantum with a Markov random walk model to explain the evolution of trust ratings over time. However the quantum vs classical dynamics is not the only difference between these models, because of the presence of the 'z' parameter in the quantum model that allows transitions between 0-10. I'm not sure why such a parameter was not included in the Markov model. However at the least it would be good to clarify for readers that the models do not differ only in the sense that the quantum one has indeterminate states.
2) The difference in goodness of fit of the models in Figure 3 is hard to discern. It might be useful to quote a numerical measure of this. RMSE would be fine given a formal model comparison is not being undertaken.
3) My impression is that no formal model fitting was carried our to find optimal model parameters. It would be good to be explicit about this if true, and perhaps indicate why.
Author Response
|
|
Reviewer 1 comments |
|
|
1 |
The paper contrasts a quantum with a Markov random walk model to explain the evolution of trust ratings over time. However the quantum vs classical dynamics is not the only difference between these models, because of the presence of the 'z' parameter in the quantum model that allows transitions between 0-10. I'm not sure why such a parameter was not included in the Markov model. However at the least it would be good to clarify for readers that the models do not differ only in the sense that the quantum one has indeterminate states. |
The z paramater is not always used. In Figure 3 A and B we show results for z=0, i.e. standard RW walk models are being compared. Only Figure 3C shows the behaviour when the z parameter used. This parameter was introduced in the quantum model to perturb the wave-like quantum dynamics to make them more uneven. As Markov dynamics are not wave-like we did not deem it useful to introduce this parameter. We have added text in Section 2.2 clarifying this, where the Hamiltonian for the quantum model is specified, and also in section 4.1.2. |
|
2 |
The difference in goodness of fit of the models in Figure 3 is hard to discern. It might be useful to quote a numerical measure of this. RMSE would be fine given a formal model comparison is not being undertaken. |
We have calculated the RMSE and added a sentence in Section 4.1, and in the Figure 3 caption. In section 5.1. we added text to clarify why RMSE does not give an adequate indication of the predictive effectiveness for either of the models. |
|
3 |
My impression is that no formal model fitting was carried out to find optimal model parameters. It would be good to be explicit about this if true, and perhaps indicate why. |
We have clarified this in Section 4.1 and added: "We did not perform any formal model fitting and comparison to optimize parameters as part of this exploratory study. Here we present modelling results visually in form of figures alongside root mean square error (RMSE) estimates of the models as numerical indicator of goodness of fit." |
Reviewer 2 Report
Comments for authors are attached. Biggest issue is that "trust" seemingly gets lost in their "exploration."
Overall, recommendation falls between major and minor revision; i went with minor.

See markup for comments.
Author Response
|
|
Reviewer 2 comments |
|
|
|
Biggest issue is that "trust" seemingly gets lost in their "exploration." |
We would like to clarify how we are referring to the concept of "trust" and "trust calibration" in this study. We consider trust as a multi-dimensional and dynamic concept. We adopt the well-known conception by Mayer et al. (1995), who define three dimensions of trust: reliability, integrity and benevolence. Our study focuses on reliability as one dimension of trust and uses reliability ratings as a proxy for trust decisions. Our experimental design uses this sense (i.e., reliability as a dimension of trust) to generate data with which to model the "trust calibration" that the human agent undergoes. We have changed some of the text in Section 1 and added a sub-heading 1.1 ("The multi-dimensional and dynamic nature of trust") to clarify this for the reader. We have also changed the title of our manuscript (as per the next comment). Hence, when we refer to "reliability" in our manuscript we are referring to it as a dimension of "trust". The "exploratory" aspect of our study refers to the modelling approaches and mixed- methods data. By this we mean, we are exploring quantum and Markov random walk models and their parameters to model the perceived trust (reliability) of the human agent while he/she interacts with the WoZ-AI agent. Additionally, we are exploring a multitude of data streams that could be useful to tune the modelling parameters, e.g., behavioural and EEG data. Mayer, R.C.; Davis, J.H.; Schoorman, F.D. An integrative model of organizational trust. Academy of management review 1995, 20, 709–734. |
|
1 |
Title: suggest replace "calibrating" with "exploring" |
We have changed the title to more clearly reflect that trust-calibration is in the human-AI interaction (rather than calibrating the modelling approaches) to: “A quantum model of trust calibration in human-AI interactions”. |
|
2 |
13: use "quantum-like" |
We have added a footnote to clarify that we use the term 'quantum' to be inclusive of the meaning 'quantum-like', for brevity and consistency throughout the paper. |
|
3 |
131 : often one that |
added |
|
4 |
132: and one that |
added |
|
5 |
140-42: good |
Thank you |
|
6 |
156: does not have a pre- |
deleted |
|
7 |
162: define "holistically" |
Rephrased to improve clarity: "enacted holistically, that is, the dimensions are inter-dependant and not formed in isolation of each other." |
|
8 |
165: since you are using "trust" in a psych sense, psychologists indicate that reliability is repeatable and that validity is truth. |
This study focuses on "reliability" as a dimension of trust. We have changed the text in the manuscript in the Introduction to clarify this and added a subheading 1.1 ("The multi-dimensional and dynamic nature of trust") to draw out these points. |
|
9 |
170: "i would want it to" is not autonomous; suggest: in a way that helps a user to achieve an agreed upon goal |
replaced with "that aligns with" |
|
10 |
179: good |
Thank you |
|
11 |
1120-121; unclear |
We have re-phrased the text: "In this study, the noisy signal is provided by a series of interactions with an AI system and the target being sought by the human agent is the decision of whether the AI system is reliable or not." |
|
12 |
1130: suggest delete "and it changes." |
retain to fit with overall sentence phrasing |
|
13 |
p. 4, 1 st sent: revise to "initially about" |
changed as suggested |
|
14 |
equation with "exp" may confuse novice readers |
changed to 'e' as suggested |
|
15 |
1161: all of the five |
added |
|
16 |
1165: delete "or" |
deleted |
|
17 |
1176: to a reliability |
added |
|
18 |
1179: delete one of the double "the the" |
deleted |
|
19 |
1231: specify what the voucher could be used for; or delete "in the form of a voucher" |
added 'non-specific gift' |
|
20 |
1240-3: incomplete phrase |
added “we selected” |
|
21 |
1251: Before citing Fig. 28, first cite Fig. 2 |
changed to "A visual overview of the experimental design is given in Figure 2, in which Figure 2B depicts the trial structure" |
|
22 |
1256: never start a sentence with a number except with words |
Changed to "A short period … (0.3s)" |
|
23 |
p. 8: grammar errors |
We thank the reviewer for their comment, and have carefully edited to correct grammar errors. |
|
24 |
1330: please state where unused data is stored, why the data was not processed for this paper, and when it will be processed |
Our first aim was to first gain insight how modeling outcomes relate to different values of initial model parameters, so as to choose reasonable boundary conditions. For this we report representative modeling outcomes of one participant in Figure 3. The other participant's data will be processed with more formal global optimization tools as one of the future milestones of this research program. We mention this in the manuscript under subsection 'Results'. |
|
25 |
p. 10-11, 13, 14, 15: minor grammatical errors |
We thank the reviewer for their comment, and have carefully edited to correct grammar errors. |
|
26 |
1392 "model fit" for which model? unclear |
We have edited to improve clarity: "To address this, we found using an initial state vector with weighted probabilities and taking into account the initial cognitive state of the participant based on their first reliability rating (Figure 3B) leads to a better model fit for both quantum and Markov models. That is, the predictions shown in Figure 3B match the observed reliability ratings better than in Figure3A. This is also reflected by the RMSE scores for the models shown in Figures 3A and B." |
|
27 |
1396-7: a claim without the analyzed results, or a citation to another reference source |
We have deleted this sentence. |
|
28 |
1398: ditto 1396-7 |
We have edited the paragraph to clarify that we are referring to the analysed results shown in Figure 3. |
|
29 |
1440-443: ditto 1396-7 |
References added. |
|
30 |
1519-527: strong claims, weak justification or weak evidence presented; that's this reviewer's reason for changing the title from "calibrating" to "exploring" |
This paragraph describes some of our future research intentions and aspirations of its applications. We acknowledge that this paragraph points to hopeful, yet informed, speculations of the possibilities this type of research could lead to. We have edited the paragraph text to make this clearer. |